# Numerical Analysis of the Circular Settling Tank

Elahe Chero[1], Mohammadamin Torabi [2*], Hamidreza Zahabi[3], Anahita ghafoorisadatieh [4], Keivan Bina[5]

[1] Department of civil Engineering, Khavaran Institute of Higher Education, Mashhad, Iran

[2] Department of Civil and Environmental Engineering, Idaho State University, Pocatello, ID, USA.

[3] Department of Civil Engineering, Institute Superior Tecnico, 1049-001 Lisbon, Portugal.

[4] Department of civil Engineering, Institute of higher education khazar Mahmudabad, Iran.

[5] Disaster Risk Reduction Advisor, Assistant Professor of Civil Engineering, Khavaran Institute of Higher Education,  Ghasem Abad, Mashhad, Iran.

Correspondence to: Mohammadamin Torabi (toramoha@isu.edu)

**Abstract.** Nowadays, settling tank's removal efficiency is one of the most crucial matters for all Water or Wastewater Treatment Plants (WTPs or WWTPs). The unit can affect WWTP performance and improve the provided effluent quality. In this paper, the geometrical aspects of a settling tank were numerically analyzed via tracer curves, the finite volume method, and Ansys-cfx software in which, the baffle depth and diameter of a settling tank were assessed. Firstly, a previous study was similarly remodeled to verify simulation results. The impact of tank depth variation has been numerically assessed where the outcomes showed that a deeper tank could raise discharge time or the Hydraulic Retention Time (HRT). Thus, extensive discharge time may result in less polluted effluent degrading more solids. However, the tank should not be too deep based on costs. Moreover, the differential effect of baffle height was analyzed and indicated that lower height is more useful to boost the HRT. An investigation of tank diameter changes also revealed that wider diameters bring about a broader HRT.

**Keywords:** Settling Tank, Tank Depth, Tank Diameter, Tracer Curve, Finite Volume Method.

## 1. Introduction

Over the past decades, Water and Wastewater Treatment Plants (WWTPs) have drawn government attention to water, especially, environmental hazards originating from grey and sewage runoff throughout urban areas. In this regard, treatment processes can be optimally designed and operated. Therefore, one of the most critical stages in WWTPs is sedimentation in settling tanks, to degrade and remove organic matters and solids. Looking at research shows that several models have simulated and analyzed the sedimentation process numerically. To simplify methods, some assumptions were effectively used to evaluate flow pattern movement, as well as solids and particles in settling tanks.

According to previous studies, mathematical models are often applied instead of analytical solutions to reach precise flow characteristics (Imam et al., 1983). Moreover, three methods are suggested to have an appropriate

description of flow pattern movement and characteristics (Kynch, 1952). Firstly, the one-dimensional model is introduced in which solids vertical movement is considered (Kynch, 1952). Secondly, the two-dimensional model is presented for vertical and horizontal solid movement. The matter which was once used to simplify the three-dimensional model (Imam et al., 1983). Ultimately, the three-dimensional model has more benefits thanks to orienting the flow pattern. Liu and Garcia developed a three-dimensional (3D) numerical model to simulate large primary settling tanks in which a tracer study was used to investigate the tank's residence time (Liu and Garcia, 2010). The model was implemented on a settling tank in Chicago in the Metropolitan Water Reclamation District of Greater Chicago (MWRDGC). Through the case study, a computational fluid dynamics (CFD) model simulated solid removal efficiencies. The results of the research model were used to establish the design basis for tank side-water depth and inlet feed-well dimensions, etc. Liu and Garcia model outcomes can be capitalized on to decrease the cost of construction via optimized settling tank.

Vahidfar et al. in 2018 investigated and modeled a rectangular settling tank in full scale by CFD method to increase efficiency. (Vahidfar et al. 2018). Zahabi et al. also in 2018 numerically investigated the geometry of rectangular reservoir to entrap sediments, and they found the optimum geometry (Zahabi et al. 2018).

There are a wide range of parameters which can effect settling tank performance. To illustrate that, the Reynolds number, flow viscosity, the type of hydraulic flow movement, and tank dimension and design are the most significant factors in the settling unit. Schamber and Larock once used the K-ε turbulence model to simulate the settling stage applying for high Reynold's number and turbulent flow (Schamber and Larock, 1983). According to the study, coarse solids with high specific weight increases the Reynold's number; therefore, this type of model is typically conducted for a settling unit. Furthermore, a study showed that the k-ε turbulence model agreed well with some experiments in a simple geometric tank (Adams and Rodi, 1990). The quality of the computations, however, deteriorates with increasing flow complexity. In fact, the effects of flow curvature are mainly applied to clarify the differences between computation and experiment, which are not a part of the standard k-ε model. Also, a mathematical model was used to predict the velocity and particles transport pattern in secondary rectangular tanks. The particle impacts called in terms of bottom current, surface return flow, and the solids concentration distribution of density stratification on the hydrodynamics were analyzed by (Zhou and Mc Corquodale, 1992). Consequently, the model was used to simulate the so-called density waterfall phenomenon in the front end of a settling tank.

It is suggested that effluent concentration changes by velocities in the withdrawal zone (Mc Corquodale and Zhou, 1993). It is also shown that there is more upward velocity in the withdrawal zone by decreasing dens-metric Froude number for a constant discharge, showing the relationship between the dens-metric Froude number, and hydraulic and solid loads. The density of the waterfall can capture large volumes of the ambient fluid in the physical and numerical models (Zhou and Vitasovic, 1992). Also, the entrainment compensating flow rate is indirectly related to the dens-metric Froude number. Furthermore, the bottom strength of the current density, the upward flow in the withdrawal zone, and the recirculation all increase as the dens-metric Froude number decrease due to entrainment into the density waterfall.

Some research also addressed an array of computational fluid dynamics (CFD) modeling in the wastewater treatment (WWT) field (Dutta et al., 2014, Daneshfaraz et al. 2016 and Zhang et al. 2016). For instance, Wicklein et al. have proposed a good modeling practice (GMP) for wastewater application and it is based on general CFD procedures (Wicklein, et al., 2016, Daneshfaraz et al. 2017).

Settling basins can be divided into two categories in terms of geometry, which are cubic and cylindrical in shape. In this regard, circular basins are better than rectangular ones, since they need less area for construction. This might increase rectangular basin hydraulic efficiency (Stamou et al., 1989). In this study, some circular basins are considered as a three-dimensional model to simulate tank geometry and stream direction. Meanwhile, continuity and momentum equations will be analyzed via the finite volume method, and the density change of the particles is ignored. Eventually, the tracer curve will be used to evaluate hydraulic efficiency in terms of basin depth, and also the tank diameter variation will be studied to assess repercussions.

## 2. Material and Methods

An increase in settling time results in tank sedimentation efficiency in which considering the appropriate size for a tank's baffle and the weir structure are two ways to improve tank efficiency. In this light, baffles may cause returning flow when flow reaches the baffle and weir structure, namely, extending the distance that flow travels to discharge from the basin tank. In this paper, the aim is to study and evaluate the Chicago basin tank which was evaluated in 2011 to analyze the basin's depth and diameter changes and its effects on effluent quality (Garcia, 2011). In this respect, tank properties are presented in table 1.

**Table 1. Properties of settling tank**

| Parameters | Unit | Dimension |
|---|---|---|
| Tank diameter | (m) | 47.24 |
| Baffle diameter | (m) | 12.8 |
| Tank depth | (m) | 3.66 |
| Baffle height | (m) | 1.52 |
| Inlet pipe diameter | (m) | 1.37 |
| Bottom slope | - | 1:12 |

The Chicago tank is capable of maintaining flow being treated into the basin by increasing retention time which happens while a weir is considered with a shorter height causing a longer distance for the flow to exist. Therefore, the mechanism triggers to provide more time for settling. On the other hand, the flow is turning when it reaches the baffle wall. In this regard, the process is going to be evaluated via the CEM-CFD model. The mesh in the model is 12 million rectangular meshes (Tetra Unstructured Mesh), where the larger and shorter bases are 10 and 2 cm, respectively. The tank which was studied by Garcia, and flow lines along with the tank mesh system are shown in Figure 1, 2, 3, and 4. It should be added that geometrical modeling was done by Ansys cfx software in the current study. K-e turbulent model also used for simulation.

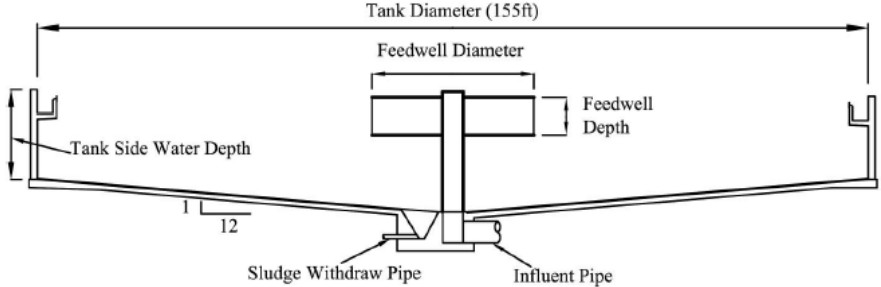

**Fig 1. Chicago tank.**

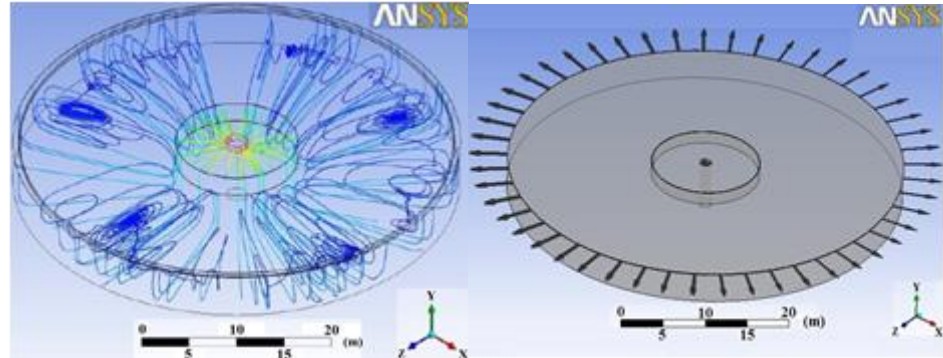

**Fig 2. Flow lines and directions in the settling tank.**

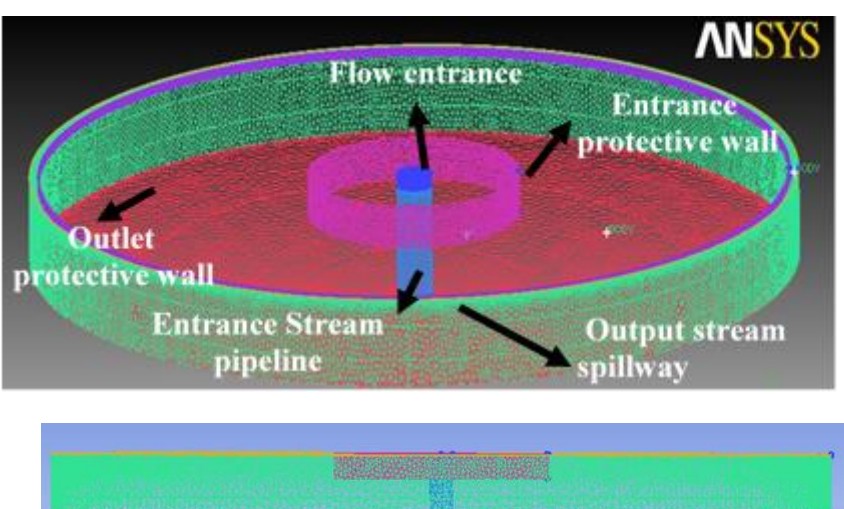

**Fig 3. Modelled settling tank.**

To simplify the model and obtain an accurate result, some assumptions are considered, including that the flow pattern is steady. Temperature variation is ignored, and flow temperature, density, and velocity are assumed to be constant (T=20 C°, Flow Density=998 Kg/m³). In addition, boundary conditions are conducted in three main terms in which the tank's surface is taken to be a slippery surface except for the bottom of the tank. The free surface is rigid and the flow pressure is calculated hydrostatically. Relative pressure at the end is zero, and the inlet is velocity radial control.

One way to calculate the settling tank's efficiency is to draw a tracer curve. The method is defined as a way in which the pigment flow is carried out to the influent and then, when the pigment reaches the effluent, the

pigment concentration is measured. Following this, three steps are taken to draw the racer curve comprised of solving the flow equation steadily in ANYSY Solver, defining the pigment in the pre-CFXANSYS, and then checking pigment concentrations in the influent and effluent after 3 hours. It should be added that hydrodynamic conditions are expressed in terms of three laws in which the conservation of mass, the conservation of momentum (Newton's Second Law) and the conservation of energy (the first law of thermodynamics) are considered.

## 3. Tracer curve method evaluation

The maximum time of the flow discharge in the current study will be compared with Garcia outcomes in the same aspect to make an evaluation (Garcia, 2011). Figure 4 shows the comparison between these two studies in the sense of tracer curves. Table 1 also shows the maximum time of the tracer curves when tank depths are taken at a 12 foot depth and two different baffle height of 2.13 and 1.52 m to compare with Garcia's results.

**Table 2. Tracer curve outcome for the two aforementioned studies**

| Tank depth (m) | Baffle Height (m) | Time of discharge (hours) (current study) | Time of discharge (hours) (Garcia, 2011) |
|---|---|---|---|
| 3.66 | 1.52 | 1.19 | 1.22 |
| 3.66 | 2.13 | 1.14 | 1.25 |

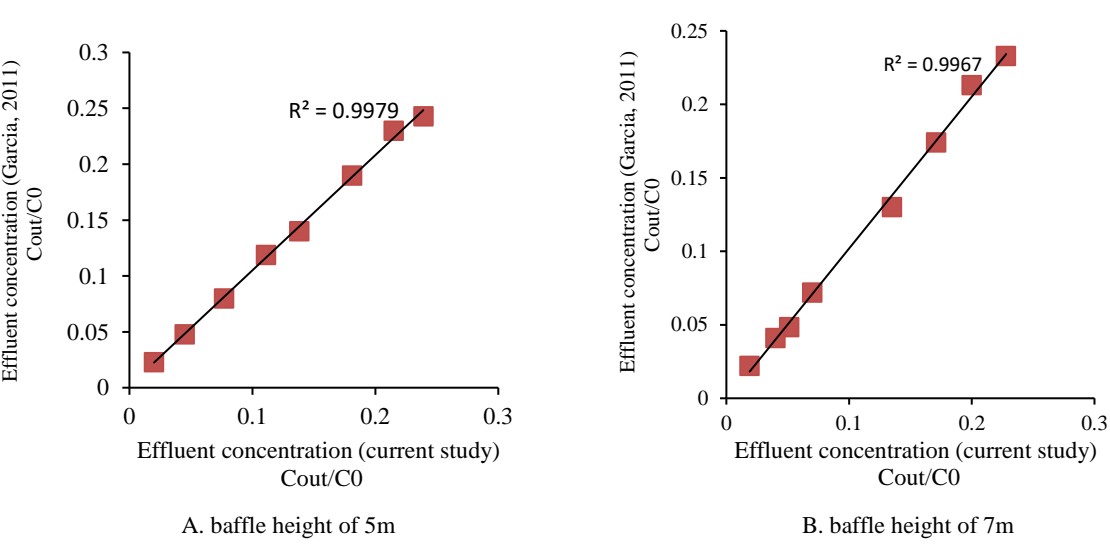

A. baffle height of 5m                    B. baffle height of 7m

**Fig 4. Data dispersion in current and Garcia studies (2011).**

As observed, data dispersion (the current study) is in good agreement with the Garcia study in which trend lines are going up by a $45^0$ slope. Beside this, the standard deviations of both A and B graphs are close to 1. Therefore, modeling of the Chicago tank by a tracer curve is effective and accurate enough to predict other basin tank depths and baffle heights.

## 4. Result and discussion

### 4.1. The effect of tank depth variation

The tracer curves evaluate the tank performance where the tank depth ($D_t$) and the baffle height ($D_f$) change with a 5-second pigment injection. Then, the pigment concentrations will be measured in the inlet and outlet (effluent) over three hours to find the difference. Figures 5 and 6 display the tracer curve results for a tank depth variation and baffle height of 1.52 and 2.13 meter, in which the tank diameter is equal to 47.24 meter.

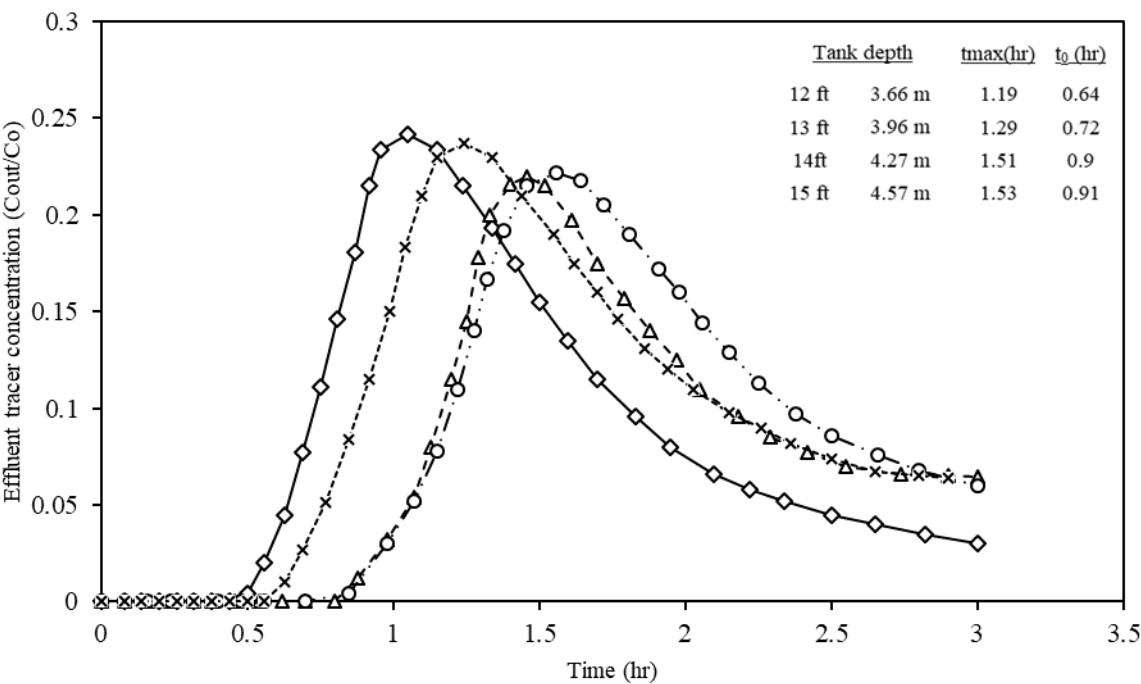

**Fig 5. Effluent concentration with a baffle height of 1.52 m in tank depth variations.**

According to Fig 5, as tank depth increases, it takes more time ($t_{max}$) to discharge effluent. Therefore, the Hydraulic Retention Time (HRT) will rise slightly which is more evident in peak point locations. It is clear from the data given that a 0.34 hr time elapse is observed from 3.66 (1.19 hr) to 1.52 m (1.53 hr) depths peak points distance. Moreover, the greater the tank depth is, the thinner the gaps between peak points become. Particularly, the gap between 4.57 and 1.22 m tank depths is narrower compared with the gap between 3.66 and 3.96 m or even the gap between tank depths of 3.96 and 4.27 m. If the tank depth is more than 4.57 m. the gap will not be noticed. Thus, tank depths which are more than 4.57 m are not economically beneficial because there would not be excessive time discharge for the tank. This means that building larger tanks is not cost efficient because it does not have a positive impact on effluent concentration.

Furthermore, the points ($t_0$) where the lines start to have more effluent concentration and the tank is getting filled with pollutions are different. To illustrate that, the starting points are 0.64 and 0.91hr, respectively, for tank depths of 3.66 and 4.57m. Therefore, deeper tanks get polluted later. Comparing the maximum points' effluent concentration indicates that the $C_{out}/C_o$ ratio falls markedly from 3.66 to 4.57m tank depths given that the optimum tank depth is 4.57m; however, there is not a significant gap between 4.27 and 4.57m depths.

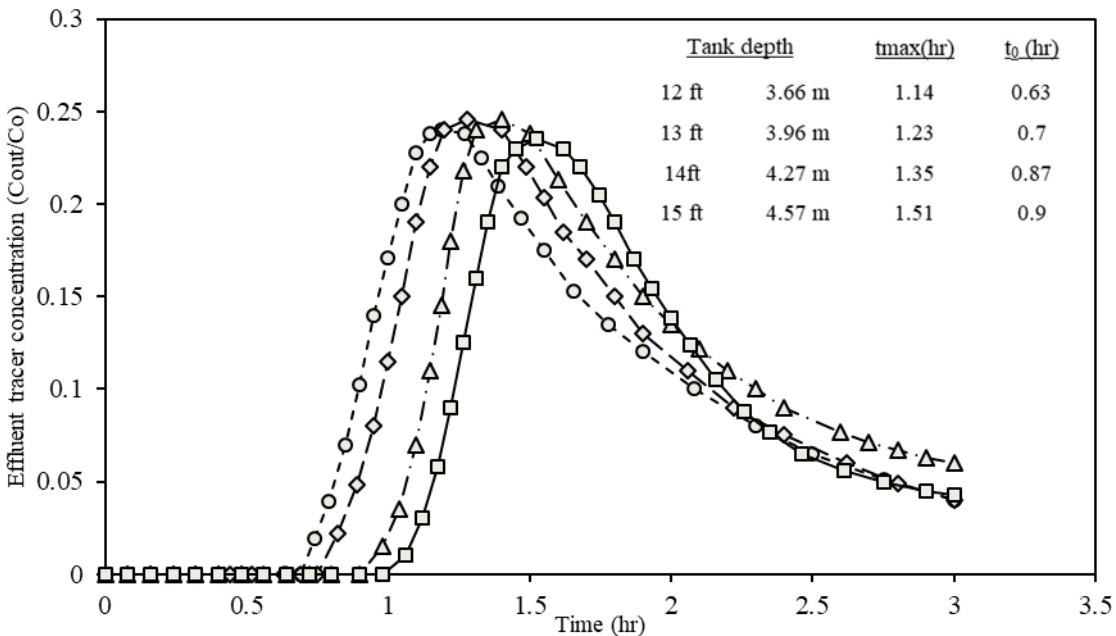

**Fig 6. Effluent concentration with a baffle height of 2.13 m in tank depth variations.**

Fig 6 (baffle height of 2.13 m) also shows a similar manner as is seen in Fig 5. Although, $t_{max}$ is slightly less than what it is in Fig 5. Plus, the effluent concentrations ($C_{out}/C_o$ ratio) is almost equal for all tank depths, with a small drop from tank depths of 3.66 to 4.57 m. Also, the same behavior holds for $t_0$ as has been discussed previously.

Overall, there is no significant difference between a tank baffle of 1.52 and 2.13m. However, a tank baffle of 5m can provide more HRT or discharge time by tracer curve calculations with the same properties.

## 4.2. The effect of tank diameter variation

Tank diameter can change $t_{max}$ and following that effluent concentration may vary. The effect of diameter variation on these parameters is analyzed in this part. A tank baffle of 1.52m generates less effluent concentration. It is selected for the following comparison. Fig 7 and 8 display tank performances for tanks that are 42.67 and 51.8m in diameter, and for which tank depths are 3.66, 3.96, 4.27, and 4.57m, respectively.

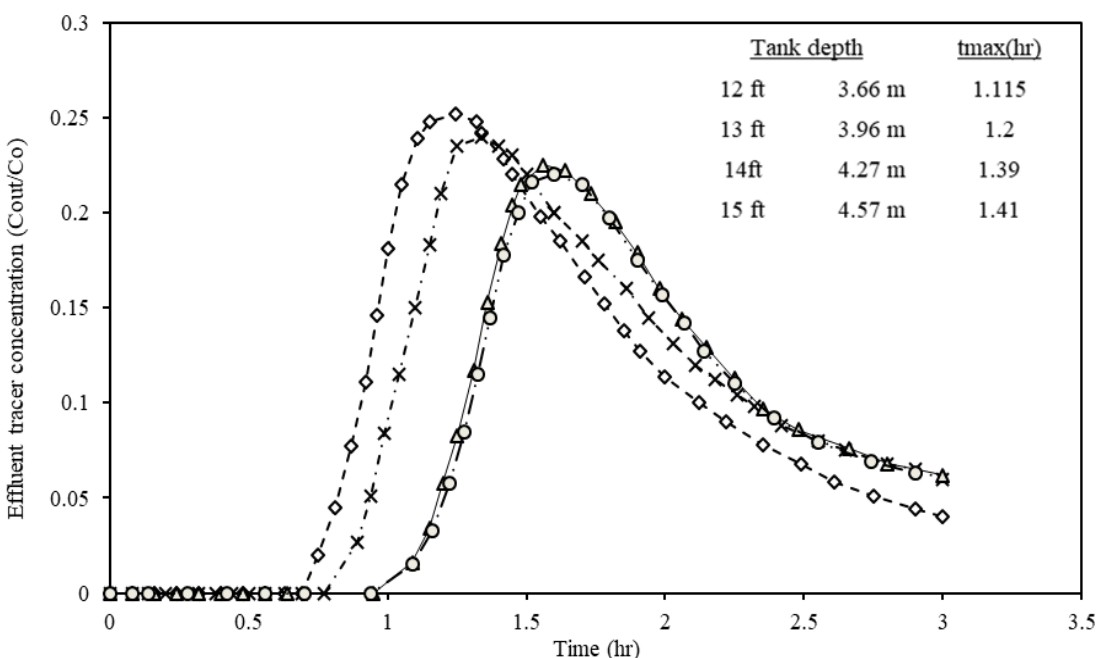

**Fig 7. Effluent concentration and tmax in tank depth variations and 42.67 m diameter.**

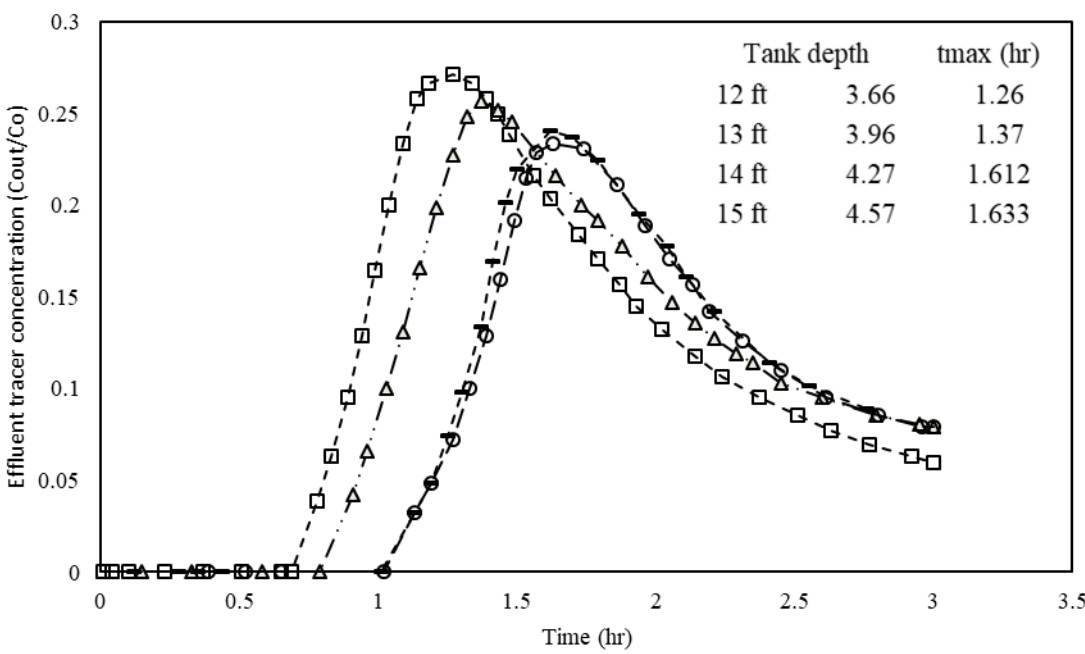

**Fig 8. Effluent concentration and tmax in tank depth variations and 51.82 m diameter.**

Fig 7 and 8 show that $t_{max}$ changes considerably when the diameter increases from 42.67 to 51.82m. $t_{max}$ rises noticeably. That is even more evident for a tank depth of 4.57m in two figures in which $t_{max}$ is 1.41 and 1.63hr for 42.67 and 51.82m diameters, respectively. Plus, there are still gaps among lines which get narrower as tank depth increases.

## 5. Conclusion

In this study, a tracer curve is used to analyze settling tank performance in which the given tank is firstly evaluated with the previous study. The results of the evaluation were homogenized with the study and similar outcomes were generated. Then, the effect of tank depth variation, baffle height, and tank diameter were investigated. It was determined that a greater tank depth increases the discharge time. Also, when the tank depth is higher, the effluent concentration is lower. Comparing baffle heights of 1.52 and 2.13m showed that the discharge time is wider with a baffle height of 1.52 m. Therefore, smaller baffle heights are effective in delaying the effluent discharge time. Tank diameter variation analysis indicated that a larger tank diameter results in a greater discharge time, which is evident for a tank depth of 51.82m compared with 45.72m. The time in which a tank gets polluted and the effluent becomes concentrated also depends on tank depth and diameter. That is more when the tank depth and diameter are considered for larger sizes.

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
