# Peer review of "Numerical Analysis of the Circular Settling Tank"

_Drinking Water Engineering and Science, 2019_

## Referee Comment (RC1) · Anonymous Referee #1 · 24 Apr 2019

I read the paper. Some of the results are original and interesting and deserve to be published. Some minor revision is due, in order to improve the robustness of the paper. Also, the topic of the manuscript is relevant to the journal and is recommended for publication provided the following concerns are addressed: #1: English should be revised to some extent; there are misprints and wrong expressions..#2: Explain more about the experiments. In general, values measured should be presented in the clearest way such as in a separate table that allows understanding the influence of them. 3#: The references should be updated. 4#: You had a turbulent flow regime so you should clearly mention which kind of turbulent model was used by the software.

---

## Referee Comment (RC2) · Anonymous Referee #2 · 26 Apr 2019

Thank you for the opportunity to review this article. The purpose, as I understand it, was to numerical analysis related to geometrical aspects of a settling tank. The study has practical implications for Civil and environmental engineering. It can be used as a usable for the wastewater treatment plan. Below, I provide specific comments and edits. These are mostly minor edits and clarifying points.

- The introduction should be shortening

- Please determine the distribution of the data (Normal etc.)

- Please use the latest references

- The manuscript needs serious English editing

- Increase the quality of the images

---

## Referee Comment (RC3) · Anonymous Referee #3 · 28 Apr 2019

Thank you for the opportunity to review the manuscript "Numerical Analysis of Circular Settling Tank" The paper outlined the numerical analysis using CEM-CFD modelling of circular settling tanks and the numerical results compared with the actual data from a wastewater treatment plant in Chicago. The numerical data and the field data, as reported in the manuscript, showed very good correlation. This indicated that the method of analyses can be reliably used to optimize settling tank size at the design stage. However, I have some concerns that in my opinion should be addressed before final publication: 1. The English grammar MUST be addresses. In many instances in the manuscript, what the authors would like to say are confused due to the poor English. 2. The units used should be SI units throughout the manuscript. 3. The measurements in the manuscript needs double-checking and correcting. Eg. line 138 'baffle height of 2.13 ft' and should be 2.13m; line 141 'tank baffle of 5 m' and what is meant is 5 ft; line 161 the word 'depth' should be diameter. 4. The conversion from 'ft'

to 'm' need correcting throughout the manuscript. In the figures the conversions must be corrected. 12 ft is given as 0.3m should be 3.65m 13 ft is gives as 0.33m should be 3.96m 14 ft is given as 0.35m should be 4.27m 15 ft is gives as 0.38m should be 4.57m 5. The introduction section should be shortened. The extensive description from the references can be summarized and shortened. 6. If possible, more details about the numerical analysis methodology can be given instead of the lengthy introduction. 7. The quality of figures 5, 6, 7, and 8 needs improving.

---

## Referee Comment (RC4) · Anonymous Referee #4 · 2 May 2019

The paper is about numerical analysis of the settling tank. Everything is well explained but there are few issues that I like to mention: - English needs to be improved - References should be updated - Some figures do not have a good quality, fix it - Check your units there is confusion between metric and Eglish system in some parts

---

## Referee Comment (RC5) · Anonymous Referee #5 · 13 May 2019

The authors present an original work that might be interesting for the DWES journal. Nevertheless, the paper should be improved to correct the grammar and also explain more the hypothesis (why a turbulent model is used and why the k-epsilon model is not sufficient) to the reader, which is not familiar with this particular problem (e.g. what Reynolds number here, etc.)
* * *

---

## Author Comment (AC1) · 1 Jun 2019

Thanks for your time and consideration, all your concern is solved in new version. Following is your comments:

- The introduction should be shortening:

Done. Introduction shortened

- Please determine the distribution of the data (Normal etc.):

I assumed you mean the effluent concentration which is almost normal distribution

- Please use the latest references:

References updated

- The manuscript needs serious English editing:

English improved

- Increase the quality of the images:

Increased
* * *

---

## Author Comment (AC2) · 2 Jun 2019

Thanks for your time and consideration, all your concern are taken care of. following is your comment

- English needs to be improved

Done, improved

- References should be updated:

Done, updated

- Some figures do not have a good quality, fix it :

Done, quality improved

- Check your units, there is confusion between metric and Eglish system in some parts

double-checked and fixed

---

## Author Comment (AC3) · 16 Jun 2019

Thanks for your comments

all your comments will be considered and implemented in the new version, for the turbulent model LES, K-e, RNG had been tried and compared and the best one chose and reported
* * *

---

## Author Comment (AC4) · 16 Jun 2019

Thanks for your detail and on point comments

All of them will apply and implemented in the new version

---

## Author Comment (AC5) · 16 Jun 2019

Thanks for your comment

Flows are laminar or turbulent and can be distinguished based on Reynolds number. Here the flow is turbulent and we need to choose between different turbulent models (k-e, LES, RNG,.. ) select one. Which has been done and compared by different model and the best one selected.

---

## Author Response (AR1)

**Dear Editor**

Thanks for your comments

Following are the comments to the Author:
"The reviewers all suggest minor revisions, and a substantial improvement of the English and proper attention to SI units. Please prepare your revision."

Response:

The English improved properly with a native editor as you can see the follows. All the units converted to metric system.

Order of the authors changed also, please consider that.

**Numerical Analysis of the Circular Settling Tank**

[revised manuscript text omitted]

A. baffle height of 5m                B. baffle height of 7m

Fig 4. Data dispersion in current and Garcia' studies (2011).

[Figure]

| Tank depth | | tmax(hr) | $t_0$ (hr) |
|---|---|---|---|
| 12 ft | 3.66 m | 1.19 | 0.64 |
| 13 ft | 3.96 m | 1.29 | 0.72 |
| 14ft | 4.27 m | 1.51 | 0.9 |
| 15 ft | 4.57 m | 1.53 | 0.91 |

**Fig 5. Effluent concentration with a baffle height of 5 feet in tank depths variations.**

[Figure]

| Tank depth | | tmax(hr) | $t_0$ (hr) |
|---|---|---|---|
| 12 ft | 3.66 m | 1.14 | 0.63 |
| 13 ft | 3.96 m | 1.23 | 0.7 |
| 14ft | 4.27 m | 1.35 | 0.87 |
| 15 ft | 4.57 m | 1.51 | 0.9 |

**Fig 6. Effluent concentration with a baffle height of 7 ft in tank depths variations.**

[Figure]

| Tank depth | | tmax(hr) |
|---|---|---|
| 12 ft | 3.66 m | 1.115 |
| 13 ft | 3.96 m | 1.2 |
| 14ft | 4.27 m | 1.39 |
| 15 ft | 4.57 m | 1.41 |

**Fig 7. Effluent concentration and tmax in tank depths variations and 140 feet diameter.**

| Tank depth | | tmax (hr) |
|---|---|---|
| 12 ft | 3.66 | 1.26 |
| 13 ft | 3.96 | 1.37 |
| 14 ft | 4.27 | 1.612 |
| 15 ft | 4.57 | 1.633 |

**Fig 8. Effluent concentration and tmax in tank depths variations and 170 feet diameter.**